# The Evaluation of a Surgical Task-Sharing Program in South Sudan



Mina Salehi [1,*], Irena Zivkovic [1], Stephanie Mayronne [2], Jean-Pierre Letoquart [3], Shahrzad Joharifard [4] and Emilie Joos [1]

1   Section of Trauma and Acute Care Surgery, Division of General Surgery, Vancouver General Hospital, 767 West 12th Avenue, Vancouver, BC V5Z 1M9, Canada
2   Responsable Médical Médecins Sans Frontières, Doctors Without Borders (OCP) Haïti, Liberia, Somaliland, Sudan & South Sudan, 40 Rector St. 16th Floor, New York, NY 10006, USA
3   Référent Chirurgie, Surgical Advisor, MSF-OCP, 34 Av. Jean Jaurès, 75019 Paris, France
4   Division of Pediatric Surgery, Department of Surgery, British Columbia Children's Hospital, 4500 Oak St., Vancouver, BC V6H 3N1, Canada
*   Correspondence: mina.salehi@vch.ca

**Abstract:** Five billion people lack access to surgery, with the highest burden being in sub-Saharan Africa. As the surgical workforce is crucial in closing this gap, the University of British Columbia collaborated with Médecins Sans Frontières to create and launch the Essential Surgical Skills (ESS) task-sharing program, which consists of online learning modules and hands-on surgical training. Our study aimed to evaluate this pilot program. This is a mixed-method prospective cohort study to evaluate the effectiveness of the ESS program in South Sudan. Quantitative data included patient outcomes (complications, re-operation, and mortality), surgical proficiency of the trainees (quiz, entrustable professional activity (EPA), and logbook data), and electronic surveys. We used semi-structured interviews to collect qualitative data. From July 2019 to February 2021, three trainees performed 385 operations. The most common procedures were skin graft (14.8%) and abscess drainage (9.6%). A total of 172 EPAs were completed, of which 136 (79%) demonstrated the independence of the trainees. During the training, surgical mortality (0.56% vs. 0.13%, $p = 0.0541$) and morbidity (17% vs. 12%, $p = 0.1767$) remained unchanged from the pretraining phase. Interviews and surveys revealed that surgical knowledge and interprofessional teamwork improved throughout the training. The program empowered trainees to develop surgical career paths and increased their local acceptance among patients and other healthcare providers. This study confirmed the feasibility of a surgical task-sharing program in South Sudan. This program evaluation will hopefully inform Ministries of Health and their partners for the development of a training pillar of National Surgical, Obstetric, and Anesthesia Plans in the sub-Saharan African region.

**Keywords:** surgical training; virtual training; medical evaluation; low- and middle-income countries; task-sharing

## 1. Introduction

Historically, surgical care and workforce development have been neglected. Today, we know that surgical diseases' disability-adjusted life years (DALYs) represent more than those of the human immunodeficiency virus (HIV), tuberculosis, and malaria combined [1]. According to the Lancet Commission on Global Surgery's report in 2015, up to 32 percent of the global burden of disease can be treated with the help of surgery [1–3]. When merging all surgical conditions, five out of seven people lack timely, affordable, and safe surgical services with adequate surgical capacity worldwide [1].

The lack of access to surgical care mainly stems from a shortage of health workers [1–4]. Surgical workforce density is the only Lancet Commission Core Surgical Indicator included in the World Bank Group's World Development Indicators, World Bank Group's 2018 Atlas

of Sustainable Development Goals, and WHO Core 100 Indicators [1,5–7]. Since surgical workforce density is mentioned in multiple developmental indicators, this signifies its weight in addressing the burden of surgical disease. For example, the maternal mortality rate decreases by almost 13 percent with every ten units of increase in the surgical workforce density [1,8]. African, Eastern Mediterranean, and South-East Asian regions account for 47% of the global population. Nevertheless, these regions only have 17% of globally licensed surgeons, according to data presented by the Global Health Observatory in 2017 [9], and these regions have approximately 41 million annual unmet surgical cases [1].

South Sudan, located in East-Central Africa, is one of the countries suffering the most from surgical workforce shortage (0.29 per 100,000 population) [4]. Due to years of civil war and conflict, a massive exodus of healthcare providers led to this country's shortage of healthcare providers. Moreover, the lack of academic training programs to replace the emigrated professionals also contributes to this situation. Currently, the South Sudanese Ministry of Health is cognizant of this gap and of the value of human resources in this matter. Their current solution to this crisis has been to select and sponsor clinical officers (COs) to be trained in surgical skills in other countries [10]. Additionally, several international non-governmental organizations (NGOs), such as Médecins Sans Frontières (MSF), have stepped up to address the unmet needs of surgical disease.

In 2019, a partnership between the MSF Operational Center Paris and the University of British Columbia Branch of Global Surgical Care (UBC-BGSC) was formalized to create a surgical task-sharing curriculum to help build capacity in MSF-supported surgical projects in sub–Saharan Africa. A multidisciplinary team of UBC-affiliated professors designed an essential surgical skills (ESS) curriculum, delivered online as eight virtual modules, with a hands-on component delivered in-field by MSF expatriate surgeon trainers. This study evaluates the implementation of this pilot program at Aweil State Hospital (ASH), South Sudan, to measure the patient and training outcomes and to identify areas for improvement in the training program. In June 2019, three medical doctors (MD) working at ASH were recruited by MSF based on their interest in surgery and their commitment to delivering surgical care locally post training. The training began on 1 July 2019.

## 2. Materials and Methods

### 2.1. Study Design

This study was a single-center, prospective observational mixed-method cohort study. The goal of this study was to evaluate the effectiveness of the training program in the surgical proficiency of trainees (quiz, entrustable professional activity [EPA], and logbook data), measure patient outcomes (complications, re-operation, and mortality), as well as the trainees' and trainers' feedback on the program. The first part was quantitative: a prospective cohort study and electronic survey; the second was qualitative: semi-structured interviews of trainees. The target population of the interviews and electronic survey was the three trainees and their in-field trainers. Ethics board approval was obtained from the UBC in Canada and the MSF in France. Care was taken to protect participants' privacy and confidentiality, and our team complied with the European Union General Data Protection Regulation (EU GDPR). Informed consent was obtained from all participants before and after their participation.

The primary study design aimed to compare patient outcomes six months before and after training. The COVID-19 pandemic delayed the training completion; thus, the post-training outcomes were measured mid-training (28 February 2021). For simplicity of reporting, we defined these outcomes as 'post-training'.

### 2.2. Tools and Data Collection

The ESS curriculum was delivered online as eight virtual modules: (1) safe surgery and safe surgical practices; (2) airway and basic anesthesia; (3) trauma resuscitation and bleeding control; (4) source control; (5) fracture management; (6) burn management; (7) pediatric considerations; (8) obstetrics. Trainees also received hands-on in-field training

by MSF expatriate surgeons trainers. The effectiveness of the training program in ensuring surgical proficiency in trainees was measured with outcome indicators on trainees' knowledge and skill performances, including pretraining and post-training test results for online modules' contents, post-module test results, number of hours spent on modules per trainee, number of completed entrustable professional activities (EPAs) per trainee, number of completed EPAs per surgical skills per trainee, number of surgeries in each category per trainee, and number of EPAs achieved per trainee. EPAs for this curriculum were adapted from the Competency-Based Medical Education (CBME) framework [11], with individual technical and non-technical skills selected based on the case mix of this particular MSF context. Each EPA was scored using the O-SCORE Entrustability scale [12], a validated 5-point Likert scale, where a score of four and above indicated successful competency in that specific EPA. The activity hours of trainees on UBC's online learning platform were collected through the UBC Canvas online learning platform.

The effectiveness of the training program in improving patient outcomes was measured by the number and types of surgeries, complication rate, re-operation rate, and mortality rate. These surgical outcome measurements were extracted from the hospital data registry and cross-referenced with trainees' logbooks.

To capture the in-field experience of trainers, we designed an online survey in which their evaluation of the training program was ranked on a Likert scale. The survey was designed on the Qualtrics platform. An email containing a hyperlink for participation was sent out to all in-field trainers.

We used semi-structured interviews to hear trainees' views on the training program. The trainees were approached through email and were invited to participate in the study. The interviews were conducted and recorded through the Zoom platform. To extract predefined themes for the survey and interview guide, we performed a scoping review of the literature on surgical training programs in low-income countries (20 December 2019) [13–18].

### 2.3. Statistical Analysis

Study data consisted of both categorical and numerical variables, for which the categorical data were presented in the form of frequency distributions and numerical data were presented in the form of descriptive statistics and comparative analysis of values attributable to the pre-training and post-training periods. A two-tailed Student's *t*-test was used to analyze monthly operative mortality rates, ICU mortality rates, and rates of operative morbidity, expressed as continuous numerical variables of mortalities/morbidities per month, divided by the monthly patient volumes, and expressed as a decimal value. A *p*-value of 0.05 was considered to be statistically significant. Qualitative data pertained to interviews conducted with the trainees during their program periods. The interview transcripts were evaluated with thematic analysis using predetermined codes to identify themes specific to each individual interview, as well as to denote common themes amongst all interviews.

## 3. Results

Three trainees participated in this first cohort of the ESS program, held over 18 months, beginning on 1 July 2019. The EPA data collection began on 1 January 2020, and procedure logbooks collection began on 1 March 2020. The surey of trainers was launched in February 2021. Four out of seven in-field trainers completed our survey. All three trainees participated in the semi-structured interviews conducted in February 2021. All data collection was completed by 28 February 2021.

### 3.1. Outcomes

3.1.1. Surgical Proficiency

The surgical knowledge test was the first formal evaluation of the training program candidates. We ran the same test with the same questions after 18 months of training.

Two of the trainees had improvements in their scores (from 78.5% and 64.2% to 85.7%). Each trainee's study time, when they were logged on to the UBC online learning platform, was tracked. This number reflected active time on the platform and did not account for reviewing downloaded materials, shared accounts, or group studying. Our trainees had log-on times from 2.5 to more than 14 h.

Each module was evaluated with a multiple-choice test. All trainees passed their post-module tests. The modules trauma management (93%, 100%, and 100%), source control (100%, 100%, and 90%), and orthopedics (100%, 93%, and 93%) had the highest test results among all trainees.

A total of 172 EPAs were completed throughout the program. The EPA scores demonstrated significant improvement over time, with the most demonstrated endpoints in January 2021 at scores of 4 and 5, indicating the competence of the trainees to perform these procedures independently. The onset of the COVID-19 pandemic resulted in a temporary shift in institutional operations, given safety precautions and trainee responsibilities extraneous to the program, reflected by the observed interruption in EPA data collection from June–September 2020.

Burn management had the most EPAs completed by trainees (76 EPAs), followed by fracture management (34 EPAs) and source control (32 EPAs). Burn management was the only module in which all trainees were evaluated for their surgical skills, followed by the source control and orthopedics modules. Module two, airway and basic anesthesia, was the only module with zero recorded EPAs (Figure 1).

NUMBER OF EPAs COMPLETED FOR EACH MODULE-PER TRAINEE

| | Safe Surgery | Airway and Anesthesia | Trauma | Source Control | Fracture Management | Burn Management | Pediatric Surgery | OBGYN | Total |
|---|---|---|---|---|---|---|---|---|---|
| Trainee 1 | 1 | 0 | 2 | 7 | 14 | 24 | 7 | 3 | 58 |
| Trainee 2 | 0 | 0 | 1 | 12 | 17 | 29 | 4 | 1 | 64 |
| Trainee 3 | 0 | 0 | 1 | 13 | 3 | 23 | 8 | 2 | 50 |

**Figure 1.** Total EPAs completed per module, represented for each trainee.

3.1.2. Patient Outcomes

Trainees demonstrated significant engagement in the program, performing 31% of all surgical procedures at the institution between 1 March 2020, and 1 January 2021 (385/1233 procedures). These values reflect MSF institutional data, which were only available until January 2021 and excluded counts of minor procedures (change of dressings). The most common procedures performed by trainees were skin graft (57 cases,

14.8%), abscess drainage (37 cases, 9.6%), wound debridement and transverse laparotomy (30 cases, 7.7% each) (Figure 2).

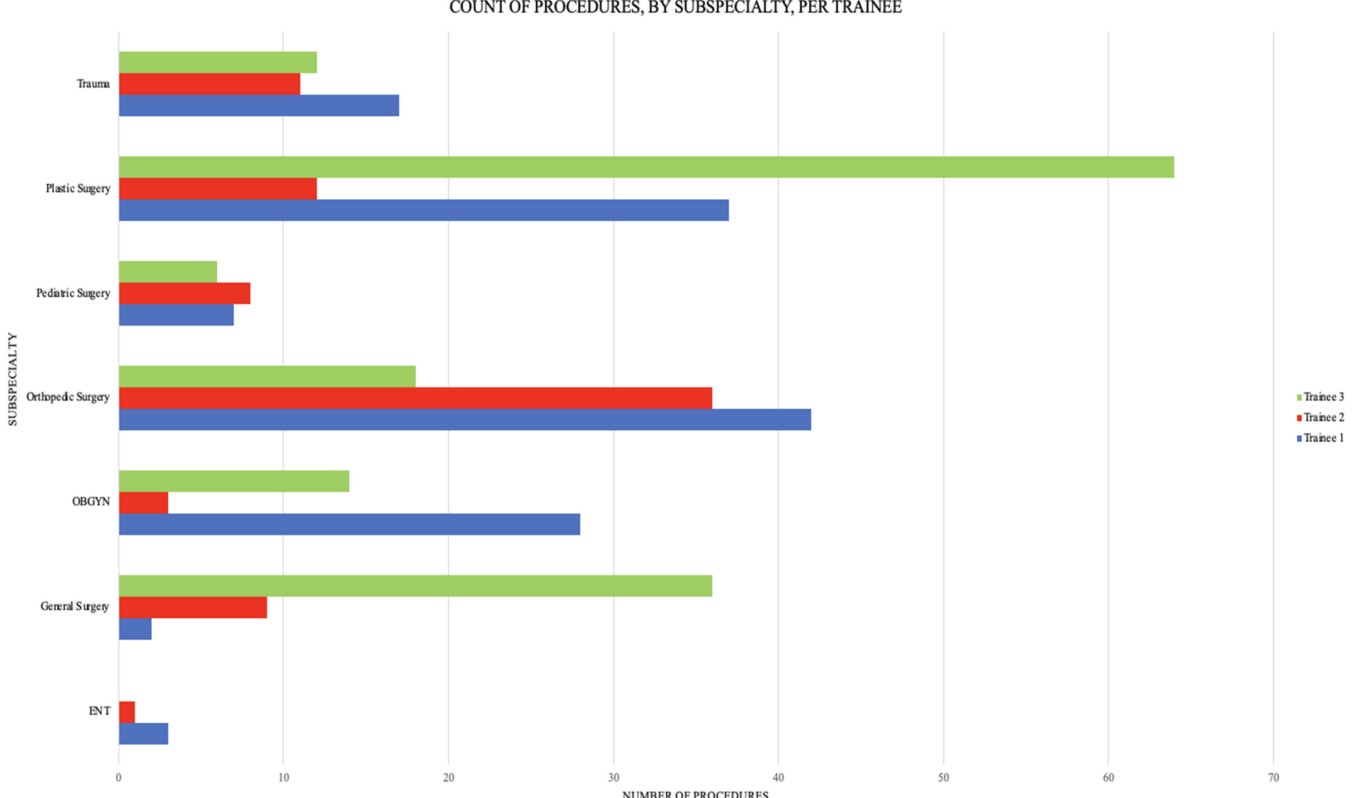

**Figure 2.** Count of procedures by type per trainee.

The surgical mortality rates recorded during the implementation of the ESS program remained unchanged from pre-implementation rates (pretraining 0.56%, post-training 0.13%, $p = 0.0541$) (Figure 3). Surgical morbidity rates demonstrated a decline from 17% pretraining to 12% post training ($p = 0.1767$), although not statistically significant (Figure 4). There was one isolated case of re-operation throughout the program implementation period due to postoperative infection. The ICU mortality rate increased during the training program (pretraining 9.25%, post training 16.28%, $p = 0.0427$). The increasing slope began in February 2020, and coincided with a decline of surgical procedures performed. We, therefore, hypothesized that this increased mortality was most probably in medical patients and an effect of the COVID-19 pandemic.

### 3.2. Semi-Structured Interviews

### 3.2.1. Empowerment/Disempowerment

All three trainees mentioned feeling empowered by this training program in performing surgical procedures confidently and felt that they had significant improvements in their surgical skills. One trainee expressed that participating in virtual workshops from Canada and being matched with a Canadian university for their training was motivating (Figure 5).

There were situations in which trainees felt disempowered. They all explained experiencing a lack of confidence in the cases they had not practiced more often due to hospital infrastructure limitations and a limited list of permitted surgeries for physicians in hospital protocol. Trainees also complained about one of the in-field trainer's unwillingness to let trainees operate independently. Furthermore, the heavy load of patients and the need for coverage of the non-surgical wards during the training program made the training experience stressful (Figure 5).

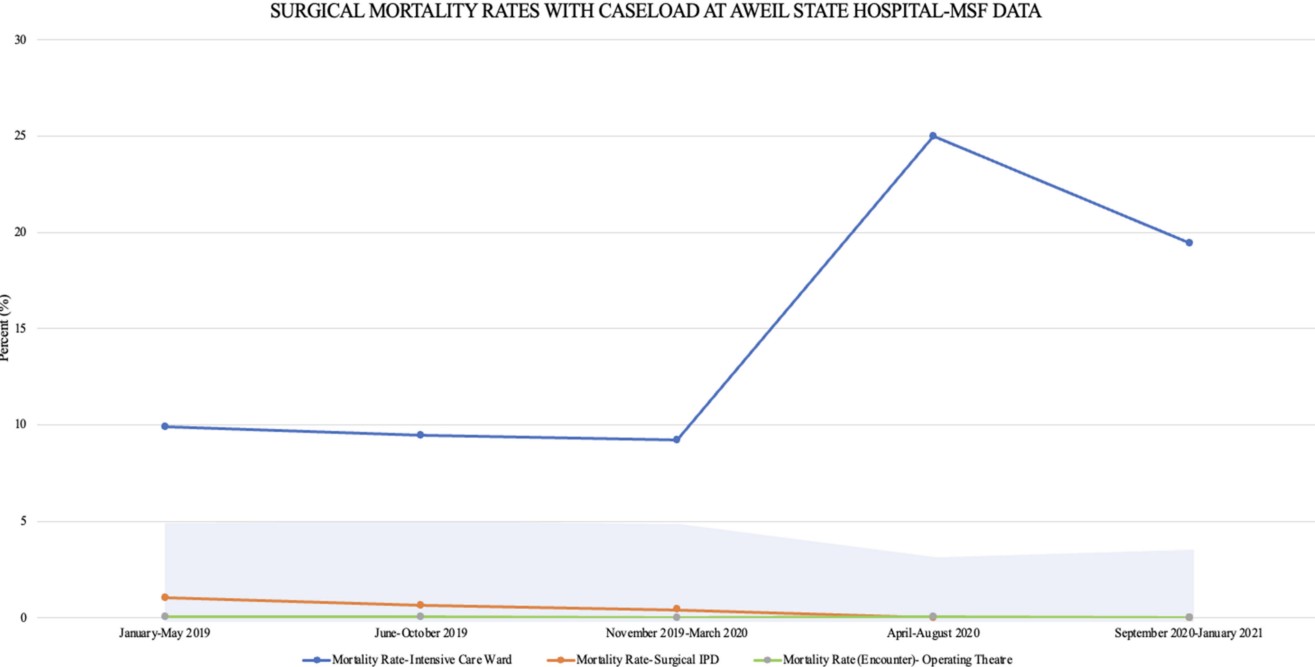

**Figure 3.** Mortality rates, represented over time as 5-month averages, superimposed with 5-month averages of surgical caseloads (% of total hospital encounters), from January 2019–January 2021 at Aweil State Hospital.

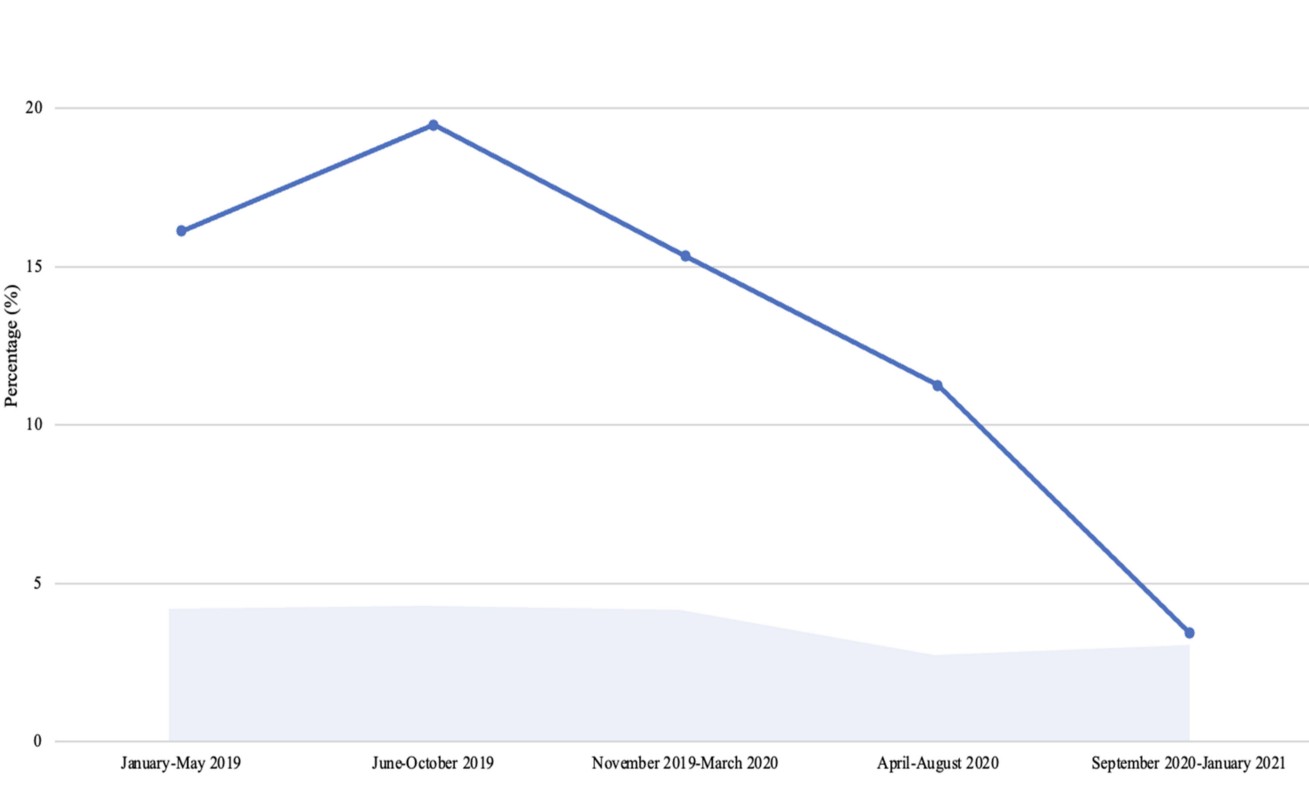

**Figure 4.** Morbidity rates, represented over time as 5-month averages, superimposed with 5-month averages of surgical caseloads (% of total hospital encounters), from January 2019–January 2021 at Aweil State Hospital.

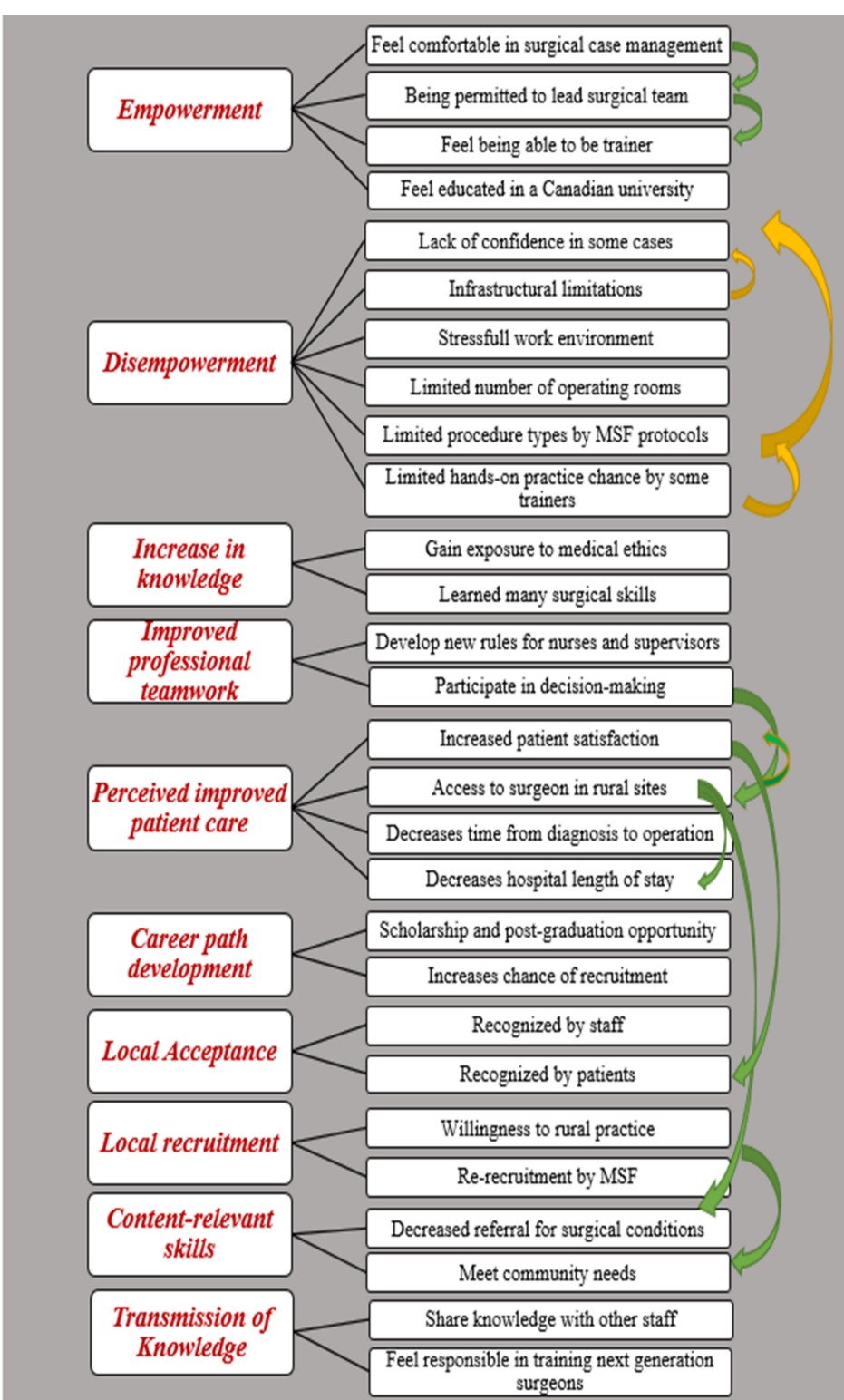

**Figure 5.** Interview mapping results. Green arrows represent the positive impact of elements on each other. The yellow arrows show the negative impact of the categories on each other.

### 3.2.2. Increase in Knowledge, Surgical Skills, and Interprofessional Teamwork

All three trainees noted that having academic training exposed them to a practice that incorporated an increased emphasis on the principles of medical ethics. Another significant area that trainees mentioned was being consulted on by other colleagues regarding surgical

management of patients, indicating the acceptance of the trainees as surgical specialists within the hospital (Figure 5).

### 3.2.3. Perceived Improved Patient Care

Every interviewee mentioned that the training program had a good reputation among patients and that the patients were satisfied with the surgical care (Figure 5).

### 3.2.4. Career Path Development, Local Acceptance, and Recruitment

Trainees mentioned the significant positive influence of this experience on their scholarship or postgraduate study opportunities. Furthermore, this training program benefited the graduates being accepted to rural hospitals that necessitated their surgical capabilities (Figure 5).

All three trainees mentioned a positive experience with their colleagues and other hospital staff as they respected the trainees as surgeons, asked for their opinion on surgical cases (even staff in other wards), and followed their medical orders. The local acceptance was not limited to the hospital cadre. All trainees mentioned being acknowledged by patients as 'surgeons' during their training process. The three trainees stated their willingness to stay in rural areas after graduation, respecting the local nature of training and the exposure of trainees to rural practice environments (Figure 5).

### 3.2.5. Content-Relevant Skills

The context of modules and hands-on skills was found to be relevant to local needs. On the one hand, overall, burn management was one of the most appreciated modules among all three trainees. They all mentioned how dramatically burn care had improved patient care, specifically skin grafting techniques. On the other hand, the orthopedics module was cited as needing more attention (Figure 5).

### 3.2.6. Transmission of Knowledge

All three trainees mentioned that they felt responsible for transferring their knowledge to their fellow physicians and nurses. One of the trainees mentioned that by spreading their knowledge, more people can benefit from these surgical skills (Figure 5).

### 3.2.7. Trainers' Survey Results

Four of the seven trainers responded to the survey (57%). From the trainers' perspectives, almost 80% (4/5) of them agreed (40%) or strongly agreed (40%) that the ability to manage surgical cases improved in the trainees over time. All trainers agreed (50%) or strongly agreed (50%) that creating a training environment at ASH improved trainees' interprofessional relationships with other staff. Moreover, 50% of trainers (2/4) agreed or strongly agreed that the assistance provided by trainees gave specialist surgeons more free time to deal with more complicated cases. Seventy-five percent of trainers (3/4) noted that they agreed or strongly agreed that the patient referral for surgical conditions had decreased, thanks to this training program.

All trainers believed that the lack of resources at the hospital disempowered trainees to perform surgeries they were trained to perform. In terms of their perspective, they mentioned that participation in a training program helped them to better understand surgical training conditions in low-resource contexts. Seventy-five percent of trainers strongly agreed with this statement, and the other 25% agreed with this matter.

## 4. Discussion

### 4.1. Feasibility of Surgical Training in Low Resource Settings

Our study showed that non-surgeon physicians (NSPs) in LMICs can benefit from an integrated training program that combines virtual didactic learning with in-field hands-on skills training to develop proficiency in ESS. All categories of the pretraining and post-training knowledge tests, post-module tests, and EPAs showed remarkable progress of

the trainees' surgical knowledge and skill performances over 18 months. The use of video recording training materials and virtual Zoom rounds with faculty at UBC also supported the trainees with reaching the required level of surgical performance and their ability to perform required procedures. The success of this approach has also been studied by Atesok et al. [19]. The Royal College of Surgeons in Ireland has taken surgical skills training one step further and shifted their Masters' students' training to a full simulation method from home. They, too, believe that virtual faculty-supervised sessions are a successful approach that can lead to increased student satisfaction [20].

The external validation of this study is limited since we studied this novel pilot surgical training program in a small healthcare facility in sub-Saharan Africa, where we were limited to three trainees. However, we believe that these findings should be disseminated for two major reasons. First, the Lancet Commission highlights the importance of partnerships between academic institutions and NGOs to augment the surgical workforce. Consequently, this effort is fully aligned with a recommendation from global surgery experts worldwide [1]. Secondly, there are very few examples of successful surgical training programs based on these partnerships in the literature. To the best of our knowledge, only four African countries offer formal surgical training to NSPs [21]. These countries are Sierra Leone, Ethiopia, Niger, and Nigeria [22]. Sierra Leone [23–25], Ethiopia [22,26–34], and Niger [22,35,36] have all begun with NGOs or high-income countries (HIC) universities supporting local training systems, creating sustainable surgical training programs in the long run. Therefore, we are convinced that this knowledge exchange could lead to the creation, development, and improvement of similar training programs across sub-Saharan Africa and other low-resource settings.

We used modified CBME frameworks to replicate evaluation formats similar to those used in HIC training programs. Due to the low-income country (LIC) setting's infrastructural limitations and the global pandemic, we acknowledge that trainees were not able to complete all modules and EPAs. During the onset of the COVID-19 pandemic, there was limited MSF expatriate surgeon availability for in-field teaching, which further contributed to the limitations in EPA completion. However, our findings showed that, with more sustained sponsorship and practical modifications such as telementoring of trainees, this hybrid model of surgical training could be feasible.

Finally, logistical constraints and trainees' dual responsibilities towards covering medical and surgical patients in the MSF setting resulted in the trainees having different exposures and case loads. The higher burn case exposure of trainees was directly correlated with the brief visit of an expatriate plastic surgeon who encouraged more aggressive burn care during his stay; hence, increasing the exposure of the trainee who was on the surgical service during that time. Unfortunately, limitations of this kind are intertwined with rural medical settings.

*4.2. Task Sharing as a Safe, Sustainable Local Solution*

The trainees significantly contributed to ASH's surgical capacity (31% of all surgical procedures). Moreover, although not statistically significant, the average surgical morbidity decreased within the study period, while the surgical department and operating theater mortality rates remained similar to the pretraining period. While ICU mortality rates were significantly higher post implementation of the training program, this timing coincided with the onset of the COVID-19 pandemic, and these rates are likely reflective of this. All these findings support the conclusion that, possibly, a surgical task-sharing program in rural areas is a safe strategy. These findings aligned with trainers' and trainees' perceptions of patient quality of care. Prior to our pilot study, the safety of task-sharing surgical training programs in low-resourced settings was evaluated and reported by Gajewski et al. in Malawi, in 2018, through a matched-pairs RCT [15] and on Zambia's surgical trainings in 16 matched-pair hospitals [37]. These studies both confirmed our findings mentioned earlier. Similar reports have been published about projects in Sierra Leone [38], Zambia [37], Guyana [17], Uganda [39], and Ethiopia [14].

The willingness of our trainees to practice in rural areas after graduation transpired through all interviews. We were not able to track the rural retention of trainees at such an early phase. However, a retrospective observational study conducted by Caleb Van Essen et al. on a 20-year program follow-up on the Pan-African Academy of Christian Surgeons showed that 100% of graduates were practicing within the Africa continent, 79% of graduates were in their home country, and 35% of graduates had more than five years rural retention [40]. Eileen S. Natuzzi et al. reported similar results in the Solomon Islands [41]. The sustainability of rural medical training was also studied by Nir Eyal and Samia A. and Parshad et al., who reported that rural training programs benefit communities [17]. Similarly, our participants agreed that the didactic and hands-on techniques included were aligned with the community's needs to various degrees. Their valuable feedback on the most necessary module modifications was recorded to be used for content improvement.

### 4.3. Strengths of the Training Program

This training program had its strengths compared to previous projects. While Gobeze et al. revealed limitations of their surgical training program in Ethiopia, including conflicts with other physicians on the decision-making hierarchy [14], the UBC-MSF ESS trainees mentioned strong connections with other physicians. They reported organically transmitting their knowledge to other physicians and non-physician staff who were not trainees in the program.

The graduates of surgical and anesthesia training in Uganda reported feelings of discomfort and disempowerment regarding the international team's ethics approach, and lack of acknowledgement for the research collaboration in a survey by Alex E. Elobu et al. [39] Our participants, on the contrary, were satisfied with their learning of new ethical approaches. As the scenarios leading to conflicts were not described in detail in previous studies, we cannot find a precise answer to the aforementioned paradox. Several factors might have also influenced the situation where there were inconsistencies between our trainees' experiences and other studies' reports. One of these factors is the colleagues' acceptance levels regarding trainees in different centers [37].

This training program also proved to be an educational environment for HIC-trained trainers, as they believed that the experience of being an in-field trainer increased their insight on surgical training in LICs, which were findings supported by Prashad et al. in Guyana [17].

### 4.4. Impact on Participants' Futures

While only three trainers mentioned the relation between trainee participation in this training program and a smoother career path, all trainees relayed that this experience was helpful for their future. As of March 2021, one of the graduates received a surgical residency position with a scholarship. A similar phenomenon was reported by Cameron et al., in 2008, in a Guyanese training program [16]. Later, in 2017, Prashad et al. updated that those participants earned promotions and participated in senior positions [17]. However, there are some potential pitfalls to this form of training. Several graduates in a training program in Zambia left for private sector practice as their advanced diploma qualification was not recognized by higher level healthcare administrations [13]. Hence, we recommend that both partner institutions, UBC and MSF, ensure that they issue a locally approved training certificate to the successful trainees. Ideally, this certificate would ensure the full integration of these graduates into the local surgical workforce.

### 4.5. Local Acceptance

Being acknowledged by colleagues and patients provides a safe educational and patient care environment. While the trainees believed they were well accepted and respected as surgeons during their training, the trainers (all HIC graduates) reported various degrees of agreement with this statement regarding trainees. The inconsistency of the feedback from trainers and trainees on this area might have been due to the two groups' different

cultural perceptions of being respected. In other words, the expectations of physicians from their colleagues and patients vary between cultures. Contrary to our findings, Guyanese trainees reported that they experienced poor organizational attitudes, which negatively affected their learning [15], such as the report from the Zambian program [13], which brings us to another area of strength in this program.

## 5. Conclusions

Since the completion of our study, our team has built a multidisciplinary team of global surgery experts and trainees—the UBC Global Surgery Lab—to translate surgical knowledge to the world through research publications, conference presentations, podcasts, talks, and stakeholders' engagement. The UBC team actively upgrades the modules, evaluation techniques, and multimedia educational tools. MSF has expressed interest in rolling out this training program in other sub-Sahara.n countries where they have surgical projects.

Having explained our findings on the surgical outcomes, educational evaluation, and perception of in-field trainers and trainees from this training program and comparing them to the previous publications brings us to the conclusion that a surgical training program in a LMIC, born out of a partnership between an HIC academic institution (UBC) and an international NGO (MSF), is a feasible, potentially sustainable, and safe approach to closing the surgical care gap in LMICs.

**Author Contributions:** M.S., I.Z., S.M., J.-P.L., S.J. and E.J. all contributed substantially to the development of this manuscript, including drafting, revising, and approving the manuscript. All authors have read and agreed to the published version of the manuscript.

**Funding:** This research received no external funding.

**Institutional Review Board Statement:** The study was conducted in accordance with the Declaration of Helsinki, and approved by the Institutional Review Board of University of British Columbia in Canada and MSF France (protocol code H21-01100 and approval date 17 May 2021).

**Informed Consent Statement:** Informed consent was obtained from all subjects involved in the study.

**Data Availability Statement:** Data is contained within the article.

**Acknowledgments:** We would like to acknowledge the contribution of the study participants and the support of the Vancouver General Hospital Trauma Services staff. We also acknowledge Morad Hameed, Naisan Garraway, Glenn Regehr, Yuwei Yang, and Rajan Bola's peer-reviewing role.

**Conflicts of Interest:** The authors declare no conflict of interest.

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
