# Peer review of "The Evaluation of a Surgical Task-Sharing Program in South Sudan"

_2673-4095, doi:10.3390/surgeries4020019_

Round 1
Reviewer 1 Report
This topic seems interesting, but some points need to be revised. Look carefully at these points:
- "This study evaluates the implementation of this pilot program at Aweil Civil Hospital (ACH), South Sudan. In June 2019, three Medical Doctors (MD) were recruited, and the training began on July 1, 2019." Materials and Is it not clear what is the purpose of this paper. Can the authors try to explain it better in the introduction section?
- "However, our findings showed that, with more sustained sponsorship and practical modifications, such as telemedicine and telementoring.." Discuss more about the role of telemedicine. Consider this very important papers: --- Telemedicine: Could it represent a new problem for spine surgeons to solve? Global Spine J. 2022 Jul;12(6):1306-1307 --- Telemedicine and Digital Health Applications in Vascular Surgery. J Clin Med. 2022
- There is no methods section. Maybe this part "Three trainees participated in this first cohort of the Essential Surgical Skills (ESS) program, held over 18 months, beginning on July 1st, 2019. EPA data collection began on January 1st, 2020, and procedure logbooks collection began on March 1st, 2020. The trainers' survey was launched in February 2021. Four out of seven in-field trainers completed our survey. All three trainees participated in the semi-structured interviews conducted in February 2021. All data collection was completed by February 28th, 2021." can be proposed as Methods section.
- Who are these three trainees? Residents, young surgeons? medical students? Explain better.
- "Figure 1. Count of Procedures by Type per Trainee." There is too much surgical procedures. Try to merge them into subspecialities.
- "Figure 5. Interview Mapping Results." It's very difficult to understand. Enhance the figure legend, adding more details.
- In the discussion section, please discuss more about the role of training in all surgical specialities: -- PMID: 34092156 -- DOI: 10.1016/j.jsurg.2023.02.011 -- DOI: 10.5435/JAAOS-D-16-00253 -- DOI: 10.1016/j.amjsurg.2020.12.055 -- DOI: 10.1080/02688697.2021.1935732 -- PMCID: PMC9953411 -- DOI: 10.3390/biomedicines11020330 -- PMID: 2895751
- Is "next steps" the conclusion of this document? It's best to write "Conclusion" and add what the authors think about the next steps to take to improve this program.
Reviewer 2 Report
Thank you for the opportunity to review this manuscript. I commend the authors for preparing this high-value article. A few of my suggestions are mentioned below:
Abstract:
1) "Mixed-method prospective cohort study" should be "A mixed-method prospective cohort study"
2) "(quiz, Entrustable Professional Activity [EPA])" - use box brackets for brackets within brackets.
3) "to02/2021" correct to "to 02/2021"
4) "(9·6%).172" to "(9·6%). 172"
5) In the Abstract, please explain the nature of the program so that the reader can understand the results in the Abstract
Introduction: Very well-written
1) The components of the program should be described in the Methods rather than the Introduction and trainees.
Methods:
1) 'Mate￾rials and Methods" should be a heading.
2) Add some details regarding the trainees. How were they selected? What was their prior experience and qualifications?
3) Have a separate subsection for "Outcomes"
4) What is a "P-test". Please note that to compare categorical outcomes such as mortality, one needs to use a Chi-squared test. Moreover, since these outcomes are paired (pre- and post-), the most appropriate test is a Mc Nemar test. Please redo the statistics for this paper and present the new p-values.
Results:
1) Please use jet black fonts in your figures. Currently it's a grey color
2) Please explain the disparities in type and volume surgeries performed amongst the three trainees. Doesn't this point towards non-standardization of the program?
3) Please use a 3-month or 5-month rolling average for the graph to make it look smoother.
4) Please explain the higher ICU mortality rates in the second half of the study period.
Discussion: "Discussion" should be a heading.
1) Can you also discuss how the model was funded and how much funding in USD was required to sustain the program.
Round 2
Reviewer 1 Report
Good
Reviewer 2 Report
Thank you for addressing my comments.